# Research Status of Mechanical Properties and Microstructure of Fiber-Reinforced Desert Sand Concrete

**DOI:** 10.3390/ma18112531

**Published:** 2025-05-27

**Authors:** Bo Nan, Jiantong Xin, Wei Yu

**Affiliations:** College of Water Conservancy, Shenyang Agricultural University, Shenyang 110866, China; nanbo@syau.edu.cn (B.N.); 2023240222@stu.syau.edu.cn (J.X.)

**Keywords:** desert sand concrete, hybrid fiber, mechanical properties, microstructure

## Abstract

This study systematically investigates the effects of the desert sand replacement ratio (DSRR) and the incorporation of individual fiber types such as steel fibers, polypropylene fibers, and basalt fibers, as well as various hybrid fiber combinations, on the workability, mechanical properties, and microstructure of fiber-reinforced desert sand concrete (FRDSC). Scanning electron microscopy (SEM) and X-ray diffraction (XRD) assessed hydration byproducts and elucidated the material’s toughening mechanisms. The optimal compressive strength occurs at 40% DSRR; further increases in the replacement ratio lead to a decline in performance. At this optimal DSRR, the addition of 0.5% steel fibers by volume results in a 27.6% increase in the compressive strength of the specimens. Moreover, the splitting tensile strength of specimens reinforced with a hybrid combination of basalt fibers and polypropylene fibers increased by 9.7% compared to those reinforced with basalt fibers alone. Microstructural observations reveal that fiber bridging promotes denser calcium silicate hydrate (C-S-H) gel development. These findings underscore the promising viability of FRDSC as a sustainable construction material, particularly for infrastructure projects in desert regions, offering both environmental and economic advantages.

## 1. Introduction

With the acceleration of urbanization, the global construction industry has witnessed an increasing consumption of natural materials for infrastructure development. Concrete, due to its ease of sourcing and preparation, dominates the building materials industry, and its demand in engineering projects remains high. Approximately two-thirds of urban buildings worldwide are constructed with reinforced concrete, with concrete comprising about one-third of the sand used [1]. For generations, river sand has served as a key fine aggregate in concrete production [2]. However, the escalating global demand for sand and gravel [3], which is increasing at a steady annual rate of 4.5%, is intensifying concerns over dwindling natural resources. Data show that large-scale construction projects may require up to 3000 tons of sand, while 30,000 tons of sand is needed per kilometer of highway. Major projects such as hydropower plants and nuclear power stations may even consume hundreds of thousands of tons of sand [4]. The extensive use of natural building materials has raised significant sustainability concerns [5,6], including the imbalanced supply–demand relationship for river sand and the prevalence of illegal sand mining. Some scholars predict [7] that global demand and prices for construction sand will continue to rise beyond 2018, as shown in Figure 1. Furthermore, sand extraction from oceans, rivers, or lakes can have severe ecological impacts [8,9,10], such as habitat destruction, the disruption of ecological communities, and increased risks of seawater intrusion leading to soil salinization and riverbank erosion, which could compromise riverbed stability [11]. Fine aggregates in concrete can be replaced by various alternatives, such as manufactured sand [12,13], sea sand [14,15], recycled concrete aggregates [16], industrial byproducts (e.g., steel slag powder, slag powder, marble powder, and granite powder) [17,18,19,20], and rubber granules [21].

The Earth has vast desert areas (approximately 6 million square kilometers), which are rich in desert sand reserves [22]. Construction sand and cement expenses in desert zones are about 50% and 10% greater, respectively, compared to non-desert locations [23]. It is well known that housing construction in desert regions is increasing daily. Extensive research has demonstrated that desert sand shares key chemical properties with river sand, containing ${\mathrm{SiO}}_{2}$ and ${\mathrm{Al}}_{2}$$O_{3}$, and has confirmed its feasibility as a concrete ingredient [24,25]. Desert sand concrete (DSC) is an innovative and environmentally friendly building material. If large desert areas could be harnessed for concrete production, it would not only alleviate the sand scarcity issue but also significantly contribute to local economic development.

Cement-based materials, such as concrete and mortar, are prone to brittle failure due to their low tensile strength. Therefore, effective measures are required to improve concrete performance [26,27,28]. Incorporating discontinuous fibers into cementitious materials has been shown to enhance crack resistance and delay crack propagation [29,30,31]. Numerous successful applications of fiber-reinforced concrete already exist. For example, the world’s first carbon-fiber-reinforced concrete structure, the Cube experimental building at the Dresden University of Technology, achieved a 50% reduction in carbon emissions during construction, representing a major advancement in sustainable building technology. In Norway, the North Cape Tunnel utilized steel-fiber-reinforced concrete to enhance stability, durability, impermeability, and crack resistance under complex geological and harsh environmental conditions, setting a benchmark for subsea tunnel construction. Similarly, Shanghai Metro Line 12 applied steel-fiber-reinforced concrete in key components, such as protective door structures, significantly improving structural strength and reliability. This application has enhanced safety and durability in the urban rail transit system, demonstrating the material’s effectiveness in large-scale infrastructure projects.

Incorporating fiber reinforcements into DSC is a promising approach to enhance its performance by leveraging the fiber’s “bridging” effect, which inhibits crack initiation and propagation. This paper provides a comprehensive review of the current research on the fundamental mechanical properties of DSC and examines the effects of commonly used fibers, such as steel fibers (SFs), polypropylene fibers (PPFs), and basalt fibers (BFs), on the mechanical performance and durability of DSC. Given the limitations of single-fiber reinforcement, the use of hybrid fiber combinations is proposed as a potential solution. Finally, the paper discusses the principles of structural enhancement and failure through microscopic analysis.

## 2. Basic Characteristic of Desert Sand (DS)

### 2.1. Physical Properties of DS

An in-depth examination of the underlying characteristics and unique properties of DS is crucial to evaluate its technical feasibility and potential benefits as an alternative material for concrete production. The DS particle morphology substantially influences the concrete’s workability. The researchers observed display the morphology and surface features of the sand particles by scanning electron microscopy (SEM). Although these sand particles show some differences in appearance, they are generally rounded and lack sharp edges [32,33], mainly due to the effect of aeolian transport in desert environments. By increasing the magnification of the microscope finer details such as disc-shaped cavities and pitted surfaces can be displayed more clearly [34]. Research has shown that DS has finer particle dimensions, superior sphericity, and enhanced surface smoothness relative to conventional river sand.

Studies have shown that DS from different regions, including the Arabian Peninsula [35], India [36], China [37,38], and Australia [39], exhibit similarities in particle characteristics, with a relatively concentrated particle size distribution and generally minor local gradation differences. Figure 2a reveals that roughly 60% of the particles measure 0.15 to 0.30 mm, surpassing the ASTM C33 maximum [40]. The non-compliance of DS gradation with the standard requirements for fine aggregates poses a major challenge to its application in concrete. To address this issue, M. N. Akhtar [41] successfully obtained a gradation that meets the ASTM C33 standard for medium sand (fineness modulus of 2.3–3.1) by blending natural desert sand (NDS) with different fineness moduli (1.2 and 1.8) and recycled crushed sand (RCS) recovered from demolished concrete specimens, as shown in Figure 2b. Research indicates that partially replacing medium sand with DS significantly improves compressive strength by 15–25% and tensile strength by 10–20% [42,43,44]. DS’s efficacy stems mainly from its minute particulate size and sleek texture, facilitating a thorough infill of concrete’s microvoids and inter-aggregate spaces. Therefore, from the standpoint of physical performance, incorporating a suitable amount of DS as fine aggregate is considered feasible.

### 2.2. Chemical Properties of DS

As shown in Table 1, the chemical compositions of different DSs are similar, the main components being ${\mathrm{SiO}}_{2}$, ${\mathrm{Al}}_{2}$$O_{3}$, and CaO, along with small amounts of other oxides. These compositions are comparable to those of river sand [45]. The sulfates and chlorides present in the desert sand can potentially lead to chemical erosion in mortar and concrete. However, E. S. Abu Seif [46] measured total dissolved salts (TDSs) in samples containing desert sand from the Saudi Arabian region and reported an average sulfate concentration of only 34 ppm, indicating that the levels of sulfates and chlorides are low enough to pose minimal risk. Similarly, as shown in Figure 3, H. Cai [45] conducted XRD analyses on desert sands from various regions and found that their mineral compositions are comparable to those of river sand. These findings suggest that the use of desert sand in concrete is unlikely to cause deterioration due to aggregate chemistry. Studies have reported [47] that DS can dissolve ${{\mathrm{Si}}_{4}}^{+}$ ions, thus facilitating the pozzolanic reaction. Furthermore, DS contains active pozzolanic substances, which allow it to participate in chemical hydration reactions [48,49]. Further research indicates that DS particles smaller than 175 µm exhibit significant pozzolanic activity and heterogeneous nucleation, providing nucleation sites for hydrates and accelerating the hydration process [39]. This characteristic contributes to an overall improvement in the strength of the material. Therefore, because of its unique reactivity, DS presents notable advantages for concrete production.

## 3. Desert Sand Concrete

### 3.1. Workability of Desert Sand Concrete

Effectiveness assessment of DSC implementation hinges on workability as a pivotal criterion. According to previous research, the processing performance of DSC meets the general requirements of engineering applications [39,55]. However, its flowability and anti-segregation capacity decrease as the DSRR increases [56,57]. T. Bouziani developed a statistical model to analyze the relationship between various factors and the flowability of concrete specimens, utilizing mini slump cone tests and V-funnel flow time tests. The model demonstrated a maximum deviation of only 7% between predicted and experimental results, confirming that increasing the dune sand replacement rate results in reduced slump flow [58,59]. The study reveals that [60] when DSRR is between 10% and 30%, its flowability improves to varying degrees. At a 20% replacement rate, the flowability of the mortar achieves optimal performance, showing a 13% increase compared to the control group. DSC demonstrates superior cohesiveness and water retention, with no observed segregation or bleeding. The slump at this point reaches a peak value of 88 mm [61]. However, DS has a relatively high water absorption capacity, which can interfere with the performance of the cementitious system, leading to a noticeable decrease in flowability and potentially resulting in a “stiff” consistency, thereby worsening workability.

Thus, determining the optimal replacement proportion of desert sand in fine aggregates is crucial. When the replacement rate is at an optimal level, desert sand significantly enhances the workability of concrete and mortar. This effect is similar to creating a “lubricating layer” between the cement and river sand, reducing internal friction and improving the handling characteristics of the mortar [49]. However, if DSRR exceeds a threshold, it expands the aggregates’ surface area and inter-aggregate voids, necessitating a greater amount of paste to cover the aggregates. This weakens the lubricating effect of the paste, increases friction between aggregates, and negatively impacts flowability during construction [62].

### 3.2. Mechanical Properties of Desert Sand Concrete

DS plays a critical role in the pore effect caused by changes in concrete porosity [63]. H. Liu [64] used ANSYS 15.0 software to establish a finite element model of Mu Us Desert sand concrete and performed a simulation of dynamic impact compression tests in the DSC. The study examined variables including specimen geometry (cylinders: 74 mm diameter × 70 mm height; cubes: 100 mm and 150 mm sides), impact velocities (5, 10, and 15 m/s), coarse aggregate content (20–60%), and particle size distribution (5–40 mm). Based on the random aggregate model to randomly generate circular aggregates in a two-dimensional plane strain model with a 2 mm mesh size, mapped and free meshing were applied to mortar and aggregates, respectively. The FEM included rigid upper and lower plates, with both mortar and aggregates modeled using the Holmquist–Johnson–Cook (HJC) constitutive model. Impact velocities were applied to the top plate, while the bottom plate was fixed. Material failure was defined as the point where the damage variable reached 1. The results indicate that DSC exhibits significant size effects. When the coarse aggregate volume is 40%, with particle sizes ranging from 5 mm to 20 mm, coupled with a DSRR of 20%, the peak stress of the concrete under dynamic loading reaches its maximum.

When using a 20% replacement rate of Mu Us Desert Sand and incorporating 10% fly ash (FA), the compressive strength peaks at 65.3 MPa, representing an 8.62% increase compared to ordinary concrete [65]. G. Zhang [66] utilized desert sand from the Tuokexun region, increasing the replacement rate to 30%. With the addition of lithium slag 20% and 1.5 kg/$m^{3}$ of PPF, the 28-day compressive strength improved by 53.26% compared to the control group. This demonstrates that moderately increasing the DSRR does not weaken concrete strength, and optimized mix designs can improve mechanical properties. As the DSRR increases, the elastic modulus and maximum deformation capacity of DSC initially rise and then decline. The results show that the stress–strain curves of DSC in elastic, elastoplastic, and yielding stages align closely with those of traditional concrete, suggesting the mechanical behavior of DSC is predictable, providing valuable reference for theoretical calculations and formula derivation [67].

Although several studies report a decline in DSC’s mechanical properties when the DSRR exceeds 50%, recent research demonstrates that optimized mix designs and novel processing methods can offset this drawback. For instance, M. N. Akhtar [68] observed that compressive strength decreases when DSRR surpasses 50%. Nevertheless, based on current concrete production practices in China, substituting 50% of river sand with desert sand could save approximately 735 million tons of river sand annually. S. M. Kazmi [69] introduced a compressive casting method to produce high-performance concrete using desert sand as a full replacement for river sand. Nine mix designs were tested (DSRRs are 0%, 50%, and 100%; strength grades are 30, 50, and 70 MPa). At 100% DSRR, compressive strength improved by 124%, 89%, and 65% across the three grades compared to non-compressed DSC. Furthermore, splitting tensile strength improved by 54%, 20%, and 12%, respectively, compared to conventionally cast river sand concrete. These improvements were attributed to enhanced pore filling and the formation of a denser microstructure. Additionally, compressed DSC demonstrated a 43% reduction in ${\mathrm{CO}}_{2}$ emissions and 42% lower energy consumption per unit strength relative to conventional concrete, underscoring its environmental benefits. F. Rahmani [20] also completely replaced river sand with desert sand in the Sahara region in self-compacting concrete and utilized granite powder—a byproduct of granite processing—as a partial cement substitute. The study assessed the effects of varying granite powder contents on the concrete’s rheological, mechanical, and durability properties. Optimal performance was achieved with a 1.7% high-range water reducer and 10% granite powder, yielding good durability under acidic and sulfate exposure while maintaining strength at elevated temperatures. Replacing 10% of cement with granite powder also reduced ${\mathrm{CO}}_{2}$ emissions by 10–11.25% per ton of cement. These innovative strategies not only address the mechanical limitations associated with high DSRR but also enhance the sustainability of DSC.

The DSC’s frost resilience is also significantly associated with the DSRR [70]. With increasing DSRR, ultrasonic velocity loss rates, mass loss rates, and DSC peak strain initially decrease and then increase, peaking at a replacement ratio of 40%. Y. Li [71] studied frost resistance and degradation mechanisms across multiple scales, finding that at a 100% replacement ratio, the strength decreases. This is because DS optimizes particle composition and porosity, suppresses water migration, and reduces permeability [72]. However, excessive DS causes aggregation, leading to strength reduction [73].

The influence of fine aggregates’ particle size distribution on coarse aggregates varies, impacting the strength of the DSC [74]. As shown in Figure 4a, when only coarse aggregates are present, significant voids remain within the concrete, reducing its overall strength. Adding appropriate amounts of river sand reduces voids in coarse aggregates, increasing compactness and improving mechanical performance. Figure 4b shows that adding moderate amounts of DS optimizes particle distribution, improves the density of the cementitious layer, and allows particles of different sizes to fill the gaps mutually. The smooth surface and small variation in the particle size of DS reduce the friction between the slurry and the aggregates during molding, lower the water demand, improve the fluidity, and improve the compactness of the concrete. However, as shown in Figure 4c, the excessive content of DS makes it the primary aggregate, leading to reduced compressive strength. This is mainly attributed to inadequate aggregate gradation, which increases voids and exacerbates the “boundary” effect. This increases the local water-to-cement ratio and reduces the structural density of the interfacial transition zone (ITZ).

Researchers have further analyzed the mechanical behavior of DSC structural components under dynamic loads. In H. Liu’s study [54], by using a Split Hopkinson pressure bar (Figure 5), dynamic impact tests were carried out on cylindrical specimens (Φ74 mm × 36 mm) under varying DSRRs, analyzing failure modes, dynamic mechanical properties, and energy absorption characteristics. The finite element analyses yielded results that closely matched the experimental outcomes. Some scholars have studied the behavior of desert sand concrete-filled steel tubular (CFST) tubes under axial compression [75]. The results revealed that, like traditional CFST columns, the tested CFST columns filled with DS exhibited high ductility, with an outward buckling failure mode corresponding to higher compressive strength. The average maximum load of plain concrete columns was lower than that of columns incorporating DS [76], suggesting the feasibility of using DS to replace conventional sand in CFST columns in desert regions.

Furthermore, traditional methods for studying concrete beams are applicable to DSC beam structures [77]. The failure modes and primary shear crack paths of conventional and DSC beams are largely consistent. For conventional beams, the first crack appears in the central bending zone under a load of approximately 80 KN; however, with a DSRR of 40%, the first crack requires a load of 90 KN. This indicates that DS can delay the formation of the first crack. At approximately 280 KN, both beam types exhibit flexural-shear failure, but beams with higher DSRR show greater deflection capacity, indicating that DS enhances beam ductility [78].

Researchers have also developed a novel block made from DS, the desert sand autoclaved aerated concrete block [79], to explore its seismic performance and provide data for theoretical calculations of wall load-bearing capacity. X. Zhang [80] conducted a quasi-static finite element analysis using ABAQUS to evaluate the seismic performance of DSC beam–column joints. The assessment was based on indicators such as hysteresis curves, skeleton curves, stiffness degradation, and ductility coefficients. In the study, a plastic damage constitutive model (Equation (Equation 1)) approximating the mechanical behavior of the DSC was introduced. The reinforcement effect was simulated by embedding truss elements representing the reinforcing bars. Concrete was modeled using three-dimensional solid elements, while reinforcement was represented by two-dimensional truss elements. Mesh generation was performed, and cyclic displacement loading was applied in accordance with relevant design codes.(1)$$\left\{\begin{array}{ccc} y & =\alpha x+\left(3-2\alpha \right)x^{2}+\left(\alpha -2\right)x^{3} & \left(0⩽x⩽1\right)\\ y & =\frac{x}{\beta {\left(x-1\right)}^{2}+x} & \left(x⩾1\right) \end{array}\right.$$α: The parameter of the curve equation of the ascending section of the concrete, α=1.1;

β: Parameters of the curve equation of the concrete descent section, β=9.

In summary, experimental investigations on key structural components such as beams, columns, and blocks have yielded preliminary results, confirming the applicability of desert sand in engineering practices. However, compared to standard samples of traditional concrete, the variety of DSC material compositions remains relatively limited.

## 4. The Effect of Single Fiber Addition on DSC Performance

Striking an equilibrium between toughness persists and concrete strength is a persistent issue in concrete due to its inherent fragility. To address this issue, fibers have been incorporated into concrete to improve its performance. The performance parameters of several commonly used fibers for concrete reinforcement are summarized in the Table 2. This study mainly investigates the impact of SF, PPF, and BF on DSC performance.

### 4.1. Steel Fibers (SFs)

In 1910, U.S. researchers initiated the evolution of steel fiber technology with their pioneering work on steel fiber-reinforced concrete (SFRC) [86]. In 1996, the American Concrete Institute (ACI) established Committee 544 on fiber-reinforced concrete, dedicated to advancing research in this field and paving the way for further developments in fiber-reinforced concrete. Since then, SFRC has remained at the forefront of concrete research, driving progress in other types of fiber-reinforced concretes.

C. Qu’s study indicates a marked rise in DSC’s compressive strength due to hooked-end steel fiber inclusion [87]. As shown in Figure 6, a comparative study was carried out with different volumes of DSRR and steel fiber (0%, 0.5%, 1.0%, 1.5%, and 2%). When the DSRR was 40% and the steel fiber volume fraction ($V_{sf}$) was 0.5%, the compressive strength peaked at 58.7 MPa, representing a 27.6% improvement over the 46 MPa control group. Moreover, even when the $V_{sf}$ was raised to 1.0%, keeping the DSRR at 20% still yielded excellent performance, with a compressive strength of 56.8 MPa, representing a 23.5% improvement over the control. These findings indicate that optimal compressive strength is attainable through precise calibration of the DSRR and steel fiber ratio.

The research also indicates the existence of an optimal range for steel fiber dosage. When the $V_{sf}$ is below 1.0%, it effectively enhances the compressive strength of concrete. However, exceeding this threshold may lead to strength reduction due to weakened interfacial transition zones between fibers and the matrix, where stress concentrations can form under load, promoting crack propagation [88]. The experimental results confirm that a DSRR of 40% yields the best overall material performance, which is consistent with the conclusions drawn by S. Lv [89]. It is also worth noting that excessively high DSRRs increase water demand, which can necessitate additional mixing water and result in construction issues such as bleeding and inadequate compaction, ultimately affecting the final performance. Moreover, steel fibers, through their unique bridging mechanism, effectively suppress the initiation and propagation of microcracks, leading to a strong positive correlation between the tensile strength of the splitting and the fiber content.

N. Kacahouh [90] examined the synergistic impact of steel fibers and desert sand within recycled concrete aggregates. The study revealed that when the desert sand content was kept constant and the recycled coarse aggregate (RCA) replacement ratio remained below 30%, the compressive strength of the concrete was largely unaffected. However, once this threshold was exceeded, the material exhibited significant changes: water absorption increased, while ultrasonic pulse velocity, volume resistivity, and abrasion resistance all declined.

In particular, the incorporation of SF effectively mitigated the adverse effects introduced by RCA. The experimental data indicated that cylindrical specimens composed entirely of RCA achieved compressive strength comparable to that of natural aggregate (NA) concrete, but only when the steel fiber content hit the 2% mark. Moreover, the incorporation of SF notably enhanced the concrete’s hardness attributes, such as its wear resistance, an outcome aligning with El-Hassan’s research [91]. Based on the experimental data, a high precision predictive model (Equation (Equation 2)) was developed to estimate the compressive strength of the cylindrical object ($f_{c}^{\prime }$) from the compressive strength of the cube ($f_{cu}$), achieving a high coefficient of determination ($R^{2}$ = 0.99), providing a reliable theoretical basis for engineering applications.(2)$$f_{c}^{\prime }=0.72f_{cu}+6.34$$

In another study, H. El-Hassan [91] applied alkali-activated slag to steel-fiber-reinforced RCA systems composed entirely of desert sand. The experimental data showed that as the proportion of RCA used for substitution climbed, the alkali-activated slag concrete’s split tensile strength ($f_{sp}$) actually waned. This decline seems to largely stem from the fact that RCA is riddled with pores, leading to a pretty flimsy bond where it meets the rest of the concrete mix—that weak ITZ, as it is known. Importantly, the inclusion of steel fibers effectively compensated for these deficiencies. When the SF volume fractions were 1% and 2%, the $f_{sp}$ values increased by 98% and 193%, respectively, when compared to the control. This peaked at 7.4 MPa, effectively offsetting the adverse effects of RCA. In addition, a multivariate predictive model (Equation (Equation 3)) was developed that incorporates the compressive strength, SF dosage, and RCA replacement ratio. With a coefficient of determination of 0.97, the model demonstrated excellent predictive accuracy, offering a valuable reference for practical applications of recycled concrete.(3)$$f_{sp}=0.55\sqrt{f_{c}^{\prime }}-0.01\mathrm{RCA}+2.13\mathrm{SF}$$

Rigid fibers generally have a positive effect on the compressive strength of concrete specimens. Moreover, SF can significantly enhance the splitting tensile strength. When concrete reinforced with steel fibers from desert sand is applied in structural elements, it has been shown that, with a desert sand content of 40% and $V_{sf}$ of 1.0%, the flexural capacity of concrete beams increases by 4% to 22% [92]. The incorporation of steel fibers not only substantially improves the crack resistance of the concrete but also reduces both the number and width of cracks.

### 4.2. Polypropylene Fibers (PPFs)

PPFs are synthetic fibers made from isotactic polypropylene. Compared to individual fibers, fiber bundles exhibit poorer performance [93]. Some researchers have added PPF and mineral admixtures into DSC to improve its susceptibility to cracking and freeze resistance [94]. Predictions from composite models show that [82] at a PPF volume fraction of 1.2% with 9.1 mm fiber length, a maximum compressive strength of 45.8 MPa and a splitting tensile strength of 3.2 MPa are achieved. H. Meng [95] found that the best effect was obtained by adding 1.0 kg/$m^{3}$ PPF, using a water–cement ratio of 0.34, 20% FA, and 30% DSRR. When compared to the control group, the polypropylene fiber effectively boosted the cracking inhibition rate up to an impressive 54% while also slicing the dry shrinkage rate down by a considerable 18.3%. This substantial improvement underscores the fiber’s pivotal role in enhancing concrete’s mechanical integrity, resistance to cracking, and its anti-shrinkage attributes.

PPF is widely used in engineering field because of its small diameter, light weight, low cost, and good self-dispersion. The fiber effectively inhibits plastic cracking, crack expansion, and matrix damage of concrete, thereby enhancing the structure’s long-term performance and durability [88,96,97]. In desert areas, the inclusion of PPF really bolsters concrete’s resilience against freeze–thaw damage. The data indicate that after enduring 150 freeze–thaw cycles, concrete incorporating PPF exhibits less mass loss and a smaller dip in compressive strength compared to its fiber-free counterpart [53]. From the mechanism point of view, during the freezing process, the PPF’s elastic modulus escalates, which is key to withstanding the forceful expansion of ice [98]. Conversely, when it melts, the modulus drops, allowing the stored energy to dissipate, thereby enhancing the concrete’s overall effectiveness and underlining its critical role in enhancing durability.

### 4.3. Basalt Fibers (BFs)

While metal fibers improve concrete performance, their susceptibility to corrosion has led some researchers to propose the use of non-metallic fibers, such as BF, in DSC [99]. Derived from natural basalt rock, BFs exhibit excellent high-temperature and chemical corrosion resistance.

When the BF content is 0.5%, the slump of both ordinary and high-strength concrete significantly decreases by 61% and 44%, respectively, compared to baseline concrete without fibers [100]. Z. Jiang [101] conducted an orthogonal experiment using Mu Us Desert Sand and BF. By performing range analysis on the compressive strength and splitting tensile strength of specimens at various curing ages, the optimal mix was identified as a 20% desert sand replacement rate and 0.1% BF content. The results indicated that the combined effect of Mu Us Desert Sand and BF enhanced the splitting tensile strength and tensile–compressive strength ratio by approximately 20% compared to ordinary concrete, while the compressive strength showed minimal variation. The impact on freeze–thaw resistance was negligible. Moreover, longer fibers were found to significantly improve the permeability resistance of DSC more effectively than shorter fibers. The research indicated that [81] incorporating 12 mm BF at concentrations of 0%, 0.1%, 0.3%, and 0.5% led to a steady decrease in slump, thereby diminishing workability. Other studies have further confirmed that BFs negatively affect the workability of DSC [102].

This is likely due to the higher water absorption and uneven distribution of BF and DS, leading to reduced workability. It was suggested that [103] the optimal length for short-cut BFs with a diameter of 16µm is 36 mm, with a volume fraction of 0.31%, which can improve the fracture modulus. When the BF content is 0.25%, the compressive strength reaches the maximum. Increasing the BF content and length can improve the flexural strength, elastic modulus, and crack resistance of concrete [104].

### 4.4. Summary

Figure 7a clearly illustrates the impact of different fiber volumes on the mechanical properties. The curves for GF, PPF, and SF follow a similar trend, with strength initially increasing and then decreasing as fiber volume increases, contributing to the improvement of compressive strength. When the $V_{sf}$ is 0.5%, the strength reaches 58.7 MPa. In contrast, the basalt fiber curve differs as BFs do not contribute to compressive strength, resulting in a 48% reduction in strength. Figure 7b shows that the splitting tensile strength of the steel fiber was significantly increased by 84.3% compared with DSC without fiber. BFs have a slight weakening effect on strength, while GFs and PPFs show a moderate improvement in strength when the volume fraction reaches 0.2% and 0.1%, respectively. Therefore, it is worth exploring the combination of BF with other fibers in further research to mitigate their negative effects.

FRDSC has attracted widespread attention due to its excellent overall compressive strength and splitting tensile strength. Studies by the aforementioned researchers indicate that the incorporation of fibers can enhance the mechanical properties of desert sand concrete, particularly when fibers and desert sand are added in optimal proportions, resulting in significantly improved mechanical performance of the specimens. However, excessively high DSRR or fiber contents may lead to insufficient paste coating, uneven fiber distribution, or weak interfacial transition zones, thereby compromising the overall performance. Therefore, optimizing the DSRR and fiber dosage is crucial for improving the comprehensive performance of FRDSC. Nonetheless, investigations into the interactions among desert sand content, regional sand characteristics, fiber content, aspect ratio, shape, and other properties remain limited, highlighting the need for further research to fully understand their combined effects.

## 5. The Effect of Hybrid Fibers on DSC Performance

Fiber-reinforced concrete containing only one type of fiber provides support within a limited crack opening range and on a single level, resulting in relatively weak reinforcement of the matrix [105]. To mitigate this deficiency, investigations have investigated incorporating diverse fiber varieties to improve concrete’s comprehensive behavior. Hybrid fiber-reinforced concrete constitutes a superior composite, crafted by integrating diverse fibers into a concrete matrix at precise ratios. The combination of different fibers can exhibit synergistic effects, where the performance surpasses the sum of individual fibers. Typical fiber combinations include hybridizing fibers of the same type but of different lengths, fibers with different diameters, fibers with different Young’s moduli [106], and hybridizing coarse fibers (steel fiber, carbon fiber, glass fiber, etc.) with microfibers (basalt fiber, polypropylene fiber, polyvinyl alcohol fiber, etc.) [107]. Proper fiber blending can achieve optimal concrete performance, a phenomenon referred to by Banthia and Sappakittipakorn [108] as “synergism”.

The ratios of various fibers in a hybrid composite can differ, which requires multiple tests to confirm their effectiveness. When the $V_{sf}$ is at 0.1% alongside a 0.3% fraction of BF, the material achieves a splitting tensile strength of 4.6 MPa. In comparison, when only 0.5% BF is introduced, the 28-day splitting tensile strength drops to 1.8 MPa. Likewise, incorporating 0.5% SF results in a 28-day splitting tensile strength of 2.5 MPa [81]. The unique interfacial mechanism between SF and FA at the microscopic level makes this hybrid system show performance enhancement; it is worth exploring in depth that the inherent high modulus of elasticity of the SF, coupled with its excellent advantages in flexural performance and crack suppression, together construct a composite system with excellent performance, and this synergistic effect far exceeds the initial expectations of the people [109].

When the fiber content is set at 0.1% with a blend BF to PPF at 1:3, the hybrid fibers notably improve the mechanical performance of the DSC material [110]. It was shown that [111] incorporating PVAF and SF at a 40% DSRR results in a 28-day compressive strength of 41.03 MPa, a tensile strength of 7.5 MPa, and a tensile strain capacity of 1.467%. These results show that hybrid fibers’ mechanical properties are superior to those of single fibers.

S. Jian [112] conducted research using 30% DS to replace river sand, with a hybrid fiber volume fraction of 2%, consisting of PPF and GF. When only one type of fiber was added, PPF generally showed better enhancement than GF. Hybrid fibers outperformed any single fiber combination. The PPF–GF–DSC composite demonstrated peak compressive and splitting tensile strength at a mix ratio of 0.1% PPF and 0.05% GF, while the highest flexural strength of 4.77 MPa was obtained with 0.1% PPF + 0.1% GF. The DSC’s dynamic elastic modulus with 0.15% PPF and 0.05% GF reached 95% [113], and after rapid freezing and thawing, it still outperformed the reference group. This suggests that hybrid fiber incorporation can significantly improve compressive, flexural, and splitting tensile strengths while enhancing toughness and freeze-thaw durability.

Y. Ma [114] mixed PVAF and PPF in different proportions and conducted tests on compressive and flexural strength for concrete made with full desert sand. The results showed that the optimal hybrid effect occurred when the PPF content was 0.5% and the PVAF content was 1.0%. The flexural strength, uniaxial tensile strength, and ductility improved by 77%, 55%, and 16% over ordinary full desert sand concrete, respectively. They also proposed a constitutive model for hybrid fiber full desert sand concrete under axial compression, named MF–DSC (mixed fiber–desert sand concrete). The relationship is described in Equation (Equation 4), and numerical simulations were performed using ABAQUS 2022 software.(4)$$\left\{\begin{array}{cc} y & =0.21x+2.58x^{2}+\left(0.21-x\right)x^{3},\\ y & =\frac{x}{8.052{\left(x-1\right)}^{1.211}+x^{0}} \end{array}\right.$$

As shown in Figure 8, the simulation curve closely matched the measured curve, demonstrating high consistency in peak stress and strain, as well as good fitting across most segments. This confirmed the suitability of the MF–DSC model for the concrete material studied.

As detailed in Table 3, the splitting tensile strength of DSC with only BF was 3.59 MPa. In contrast, hybrid GF and PPF increased the splitting tensile strength to 4.38 MPa, representing a 22% improvement. Similarly, hybrid BF and PPF attained a splitting tensile strength of 3.94 MPa, marking a 9.7% enhancement. The axial compressive peak strain of DSC with hybrid PVAF and PPF was 6.27 × $10^{-3}$ [114], approximately 1.44 times higher than that of DSC with 100% desert sand replacement and no fiber reinforcement 4.35 × $10^{-3}$ [115]. Additionally, it was about 2.8 times higher than the DSC containing 60% desert sand and 15% modified rubber aggregate 2.24 × $10^{-3}$ [116]. The contribution of fiber reinforcement to the enhancement of DSC performance cannot be ignored, which has been fully verified in many studies, and its effects are worth exploring in depth.

However, when PVAF and PPF are applied to DSC, the compressive strength is not as good as it should be, and the experimental data show that the value is only 26.77 MPa [114]. This is primarily due to the high proportion of desert sand within the composite. Additionally, fiber inclusion has unintentionally heightened internal sample defects, resulting in reduced compressive strength. In a similar vein, Dawood and Jaber [118] examined how varying lengths of steel fibers affected concrete performance, testing both individual and combined methods. The research results showed that the concrete blend containing hybrid SF along with 40% desert sand had the strongest compressive strength, measuring 26% higher than the standard sample. However, when the DSRR was boosted to 100%, there was a considerable decrease in compressive strength, dipping by a notable 39%. This suggests that when working with hybrid fiber desert sand concrete, it is crucial to carefully consider the ratio of desert sand in the mixture.

## 6. Microstructure and Mechanisms of FRDSC

### 6.1. XRD Analysis

Researchers employed XRD techniques to examine core samples from various concrete mixtures, aiming to investigate the microstructural characteristics and elemental composition of cured concrete [119]. As shown in Table 4, energy-dispersive spectroscopy (EDS) results indicate that the primary elements present in all samples are calcium (Ca), silicon (Si), magnesium (Mg), aluminum (Al), iron (Fe), and oxygen (O). XRD analysis further reveals that these elements predominantly exist in the form of oxides.

In the study by Y. Huang [120], a comparison of the XRD patterns of desert sand concrete after 28 days of curing and after undergoing 90 cycles of sulfate-induced wetting and drying reveals the key factors influencing concrete durability. As illustrated in Figure 9a, well-defined diffraction peaks are observed for key crystalline phases, including quartz (${\mathrm{SiO}}_{2}$), C-S-H gel, calcite (${\mathrm{CaCO}}_{3}$), portlandite [${\mathrm{Ca(OH)}}_{2}$], and ettringite, which collectively constitute the backbone of concrete’s microstructure and govern its bulk mechanical properties. After sulfate attack (Figure 9b), all other diffraction peaks remained stable, while the characteristic peak of ${\mathrm{Ca(OH)}}_{2}$ completely disappeared. This phenomenon is attributed to the reaction between ${\mathrm{Ca}}^{2+}$ and ${{\mathrm{SO}}_{4}}^{2-}$ during sulfate attack, forming gypsum, which not only depletes ${\mathrm{Ca(OH)}}_{2}$ but also increases the concrete’s porosity. Furthermore, the newly formed gypsum may react with hydration products to eventually produce a more soluble calcite phase, thereby accelerating the deterioration of the concrete.

The abrasion resistance of concrete refers to its ability to withstand mechanical friction, impact, or wear on the surface and is one of the key indicators for evaluating its durability. FA is a fine mineral admixture produced during coal combustion in thermal power plants, while silica fume is an ultrafine powder with high pozzolanic activity generated during the smelting of ferrosilicon or industrial silicon. The study by X. Cai [121] demonstrated that the addition of FA in appropriate amounts (not exceeding 40%), combined with silica fume, can effectively enhance resistance to concrete abrasion. Considering frequent sandstorms and severe vegetation degradation in desert regions, concrete structures are particularly susceptible to erosion and cavitation damage caused by airborne particles, making abrasion resistance a critical performance attribute in such environments. Experimental research by A. Cao [122] indicated that the abrasion resistance of concrete samples increased by 80.19% and 81.59% when incorporating 10% FA or 0.05% BF individually, respectively, while the addition of 10% silica fume (SiF) resulted in a 12.50% increase in compressive strength. In contrast, DS alone reduced wear resistance. However, an optimized multi-component blend (10% FA + 10% SiF + 40% DS + 0.05% BF) improved wear resistance by 112.95% compared to conventional concrete and reduced the wear rate by 48.83%. XRD analysis of concrete containing various admixtures was conducted over a 2θ range of $10^{\circ }$–$60^{\circ }$, and the results are shown in Figure 10. Under singular-variable testing, diverse admixtures had no impact on the primary crystalline phases of the hydration products. The XRD analysis of concrete incorporating DS revealed a prominent ${\mathrm{Al}}_{2}$$O_{3}$ diffraction peak at 2θ = 45.813°, demonstrating that ${\mathrm{Al}}_{2}$$O_{3}$ actively reacts with the ${\mathrm{Ca(OH)}}_{2}$ generated during cement hydration. The formation of calcium aluminate hydrates (CAHs) improves the composite’s bulk strength properties. This reaction not only improves the compressive strength but also reduces the availability of ${\mathrm{Ca(OH)}}_{2}$ for harmful reactions with aggressive agents such as sulfates, thus significantly improving concrete durability. Moreover, the spectral analysis reveals distinct diffraction peaks corresponding to ${\mathrm{NaAlSi}}_{3}$$O_{8}$ and ${\mathrm{KAlSi}}_{3}$$O_{8}$, indicating the presence of aluminosilicate components that play a crucial role in the cement hydration process. These compounds contribute to the formation of strength-enhancing hydrated gels, specifically sodium aluminosilicate hydrate and calcium aluminosilicate hydrate [123,124], which improve the mechanical strength of the concrete.

Notably, basalt fibers do not chemically participate in cement hydration. However, XRD analysis revealed broadening of diffraction peaks associated with certain mineral phases, such as ${\mathrm{Ca(OH)}}_{2}$, in the presence of fibers. This broadening suggests a reduction in the crystal size and degree of crystallinity, favorable microstructural changes to improve the strength of the concrete. This effect is primarily attributed to the three-dimensional, randomly oriented distribution of fibers within the concrete matrix, which physically restricts crystal growth and alters crystallization behavior by limiting the space available for ${\mathrm{Ca(OH)}}_{2}$ crystal formation.

The unique particle characteristics of desert sand significantly influence the internal structure of concrete, particularly in terms of pore morphology, crack development, and void distribution. As shown in Figure 11, a comparison of silicon and calcium concentrations across different samples reveals a clear trend: increasing desert sand content corresponds to a higher silicon weight percentage, indicating a direct correlation between silicon content and desert sand proportion.

### 6.2. Fiber Microstructure

The main hydration products of cement are C-S-H gel, which serves as the backbone of cement’s structural integrity and significantly influences the characteristics of the cement paste. This gel is created via chemical reactions among cement components and water as hydration progresses. Within the paste’s microstructure, calcium carbonate phases like needle-like calcite and aragonite are often visible in microcracks and areas of fracture. The creation of C-S-H gel plays a critical role in determining concrete’s structural strength.

Figure 12a–d show SEM images of the hydration products of the glass fiber desert sand concrete (GFDSC) made with ordinary Portland cement (PO· 42.5), glass fiber with a content of 0.05–0.2%, and desert sand with a substitution rate of 10–40%. The ${\mathrm{CaSiO}}_{3}$, C-S-H, and ${\mathrm{Ca(OH)}}_{2}$ (CH) crystals formed during cement hydration displayed a fibrous, honeycomb-like morphology at 10,000× magnification. The C-S-H gel, through its extensive bonding and adhesive properties, not only surrounds crystals such as CH but also forms strong bonds with the discrete aggregates, cement components, and various hydration products in the concrete matrix, thereby creating a dense and interconnected spatial network that ensures the overall structural strength of the concrete. However, as the concrete undergoes shrinkage, hardening, and external loading, the relatively brittle transition zone at the interface gradually develops numerous fine cracks and voids. These inherent defects damage the tightly coordinated force system within the concrete, leading to a decline in overall mechanical properties. Incorporating glass fiber notably improved the cement’s microstructure density and uniformity.

In Figure 13a,b, the microstructural morphology of GFDSC under the influence of glass fibers can be clearly observed. GFs have a large cross-sectional area and length, with some protrusions on the surface, which can absorb a significant amount of hydration products. Due to the excellent insulating properties of GF, some SEM images show a darker visual effect. In the GFDSC paste, dense and uniform C-S-H crystals formed, interspersed with small granular CH crystals. The bridging effect of GF in the interface transition zone effectively suppresses the propagation of microcracks. This bridging effect not only enhances the bonding strength at the interface but also improves the concrete’s crack resistance. Furthermore, the introduction of glass fibers altered the formation process of C-S-H gel, promoting a more uniform distribution of hydration products, thereby reducing pores and cracks caused by uneven hydration. The addition of GF played a dual role, bolstering the stability of the concrete’s microstructure while simultaneously boosting its toughness—a key factor in elevating the overall performance of DSC.

### 6.3. Reinforcement Mechanism

Cracks and pores in concrete typically arise from two critical stages: first, during mixing when uniformity is not achieved; second, during the curing phase, when defects may still exist. Figure 14 [84] shows a schematic diagram of ordinary DSC after 300× magnification, revealing a visibly loose internal structure with weak bonding between cement and fine aggregates, and evident pores and microcracks [125]. As the DSRR increases, the porosity of the specimens decreases significantly. However, when the desert sand content becomes excessive, the surplus sand tends to trap air bubbles [126], which may lead to crack formation, compromise structural integrity, and reduce compressive strength.

The purpose of adding fibers to brittle concrete is to allow the fibers to share the internal stresses of the concrete, thereby slowing down or preventing the formation and propagation of microcracks. This helps to slow the rate of crack propagation. If cracks have already formed in the concrete and come into contact with fibers, the cracks will tend to bypass the fibers when the strength of the fibers is sufficient to resist further crack propagation. This phenomenon results in an increase in the area covered by cracks, significantly increasing the energy required to reach structural failure [127].

Compared with traditional DSC, FRDSC exhibits a clear distinction in the characteristics of microcracks and pores. Specifically, in FRDSC, the number of pores and cracks near them is relatively smaller. Under load, the number of cracks in ordinary DSC increases significantly, and as the load increases, the depth of cracks continues to expand, eventually leading to the penetration of hydration products into coarse aggregates, forming observable macrocracks. These cracks are usually concentrated in the weaker areas where the aggregates and paste meet. In FRDSC, microcracks are typically found at the weak interface between fibers and mortar.

Under external force, the fibers and the matrix jointly bear the load, and internal stresses and energy accumulate. Once a certain threshold is reached, cracks will emerge on the surface of these weak zones. The fibers gradually exhibit a bridging effect, and as the cracks propagate through the fibers, the load is evenly distributed to other parts via good frictional forces. Once crack propagation is restrained and energy accumulates to a critical point, cracks will re-emerge at the next weak link in the interface, and this cycle continues until the fibers eventually break.

The SEM micrograph shown below is magnified 2000 times. Figure 15a shows the SEM result without fibers, where crack width reaches up to 0.63 μm. In contrast, Figure 15b, taken in the presence of glass fibers, shows a notable reduction in crack width, approximately 11.36%.

The incorporation of fibers significantly improves the hydration environment by enhancing water retention capacity. At microscale, the uniform fiber distribution allows for close integration with cement paste and hydration products, forming a cohesive and synergistic structure. This structure achieves three primary optimizations: (1) effective suppression of aggregate settlement; (2) reduction in the local water-to-cement ratio around aggregates; and (3) minimization of weak interfacial zones. These microstructural enhancements, at the macroscale, substantially reduce the risk of fiber breakage or debonding under external stress and promote a more efficient and stable hydration process.

## 7. Conclusions

This review summarizes recent advancements in the application of FRDSC. To address the shortage of river sand, reduce concrete material costs and carbon footprints, and fully harness the potential value of desert sand, it also examines the role of fibers in enhancing the performance of DSC. The main conclusions are as follows:

1. Characteristics of Desert Sand and Its Impact on DSC: The characteristics of desert sand vary across regions. Physically, it is nearly spherical, with a small particle size and good rounding; chemically, it has a diverse composition, with significant regional differences. These properties make DSC highly flowable but also increase its water demand. DS serves to fill gaps, optimize porosity, and improve particle grading. When the DSRR is around 40%, it can enhance the strength of DSC; however, excessive replacement leads to increased porosity, reducing both strength and durability.

2. Effects of Fibers on DSC: With the addition of fibers, the DSRR should still be maintained at around 40%. In polypropylene-fiber-reinforced desert sand concrete, an inclusion of 1.0 kg/$m^{3}$ of polypropylene significantly improves durability. In steel fiber-reinforced desert sand concrete, the split tensile strength is positively correlated with the fiber volume. When basalt fibers are used in combination with other fibers, the split tensile strength increases by 9.7% compared to using basalt fibers alone. Overall, mixed fiber reinforcement in DSC shows more comprehensive improvement compared to the use of individual fibers.

3. Microscopic Enhancement Mechanism of GF-Reinforced DSC: A microscopic study at 10,000× magnification of GF-reinforced DSC reveals that fiber toughening occurs in several ways. These include stress transfer, crack propagation inhibition, increased energy consumption during failure, optimization of micro-porosity and crack characteristics, enhanced water retention, and improved stability of the cooperative stress system, which together strengthen the microstructure of the concrete on multiple levels.

This review aims to enhance understanding of FRDSC products. It highlights the potential of fibers and desert sand, facilitating the large-scale sustainable application of FRDSC and contributing to green innovation in the construction industry.

## Figures and Tables

**Figure 1 materials-18-02531-f001:**
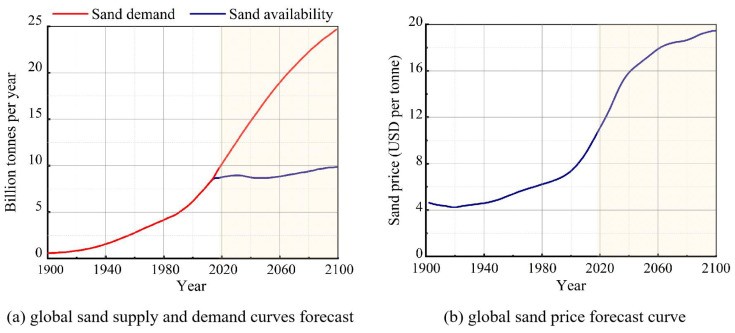
Relationship between supply and demand and price of sand for construction.

**Figure 2 materials-18-02531-f002:**
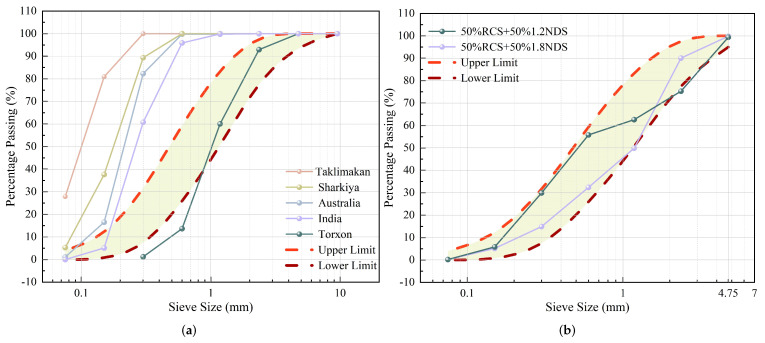
Size distribution profile of desert sand particles: (**a**) particle size distribution of DS in different regions; (**b**) particle size distribution of modified DS.

**Figure 3 materials-18-02531-f003:**
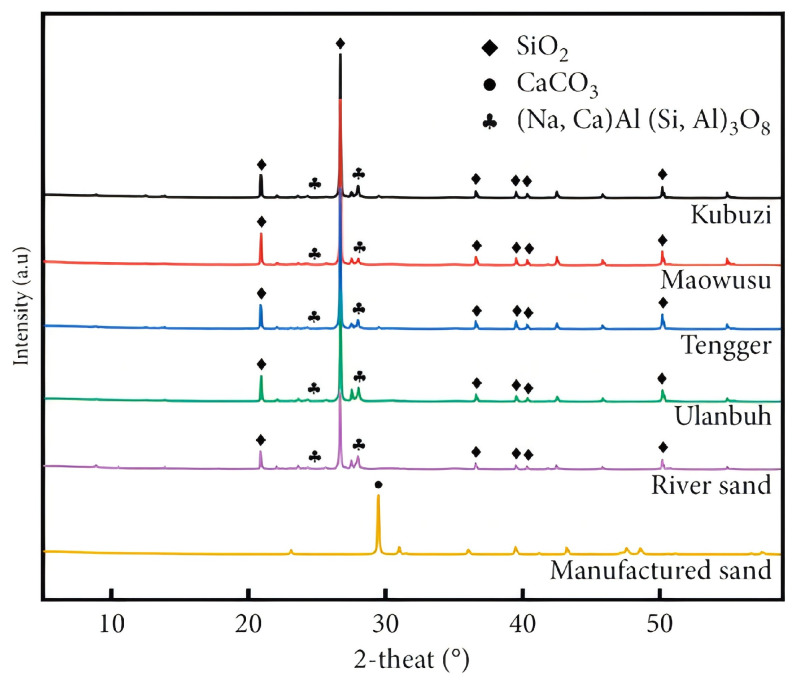
XRD of desert sand in different regions [45].

**Figure 4 materials-18-02531-f004:**
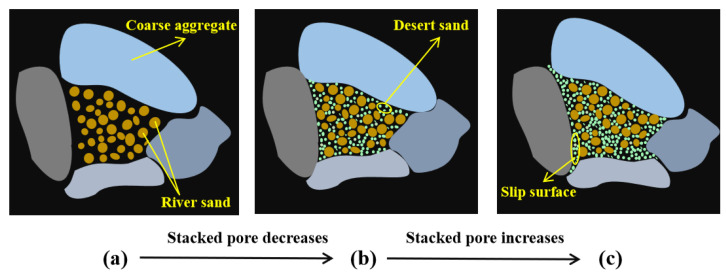
Schematic diagram of strength enhancement and weakening of desert sand: (**a**) river sand filled voids; (**b**) river sand and moderate amount of desert sand filled voids; (**c**) excess desert sand filled voids.

**Figure 5 materials-18-02531-f005:**
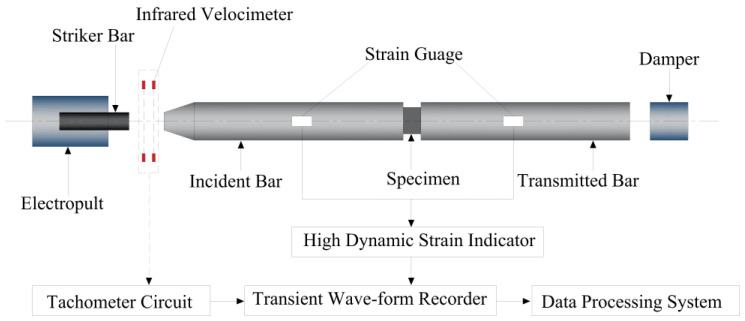
Split Hopkinson pressure bar diagram.

**Figure 6 materials-18-02531-f006:**
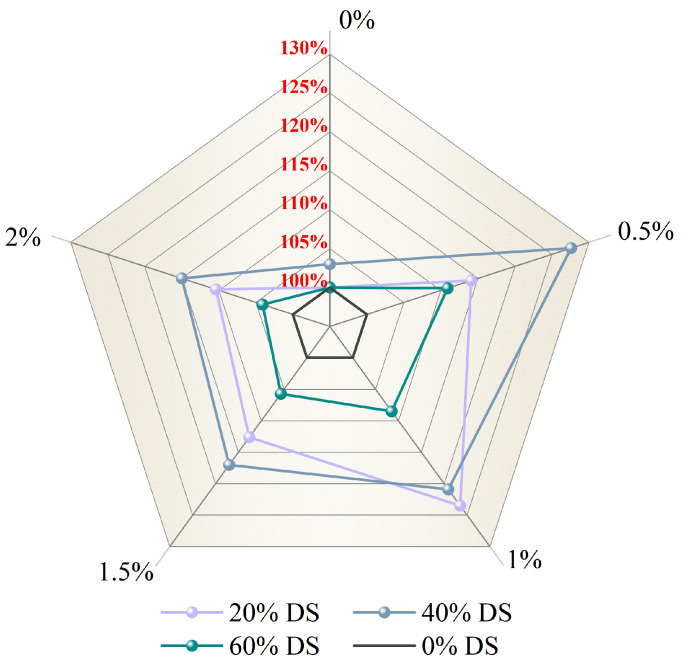
Effect of steel fiber volume fraction and desert sand replacement ratio on the compressive strength of concrete.

**Figure 7 materials-18-02531-f007:**
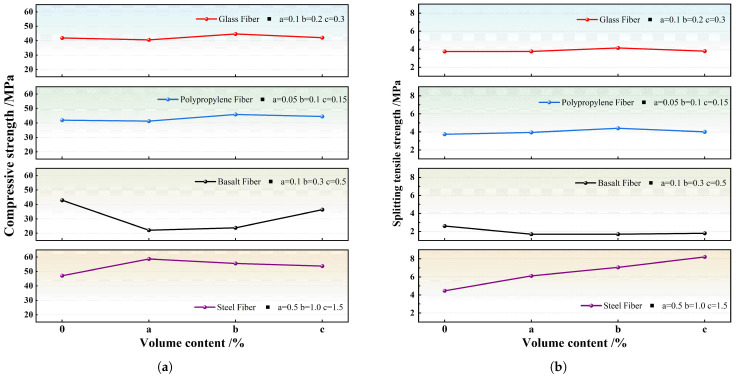
Influence of fibers on the 28-day compressive strength and splitting tensile strength of DSC [81,84,87] (**a**) Compressive strength; (**b**) splitting tensile strength.

**Figure 8 materials-18-02531-f008:**
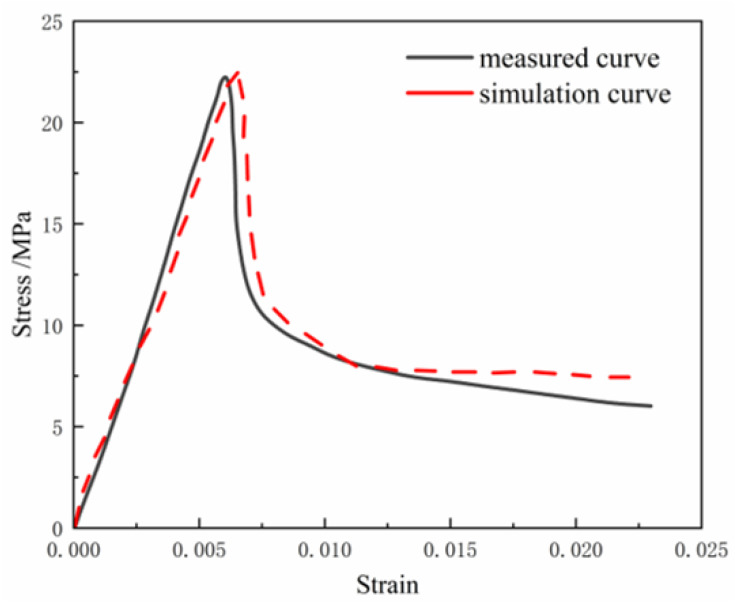
Comparison of simulation and measured curves: data source [114].

**Figure 9 materials-18-02531-f009:**
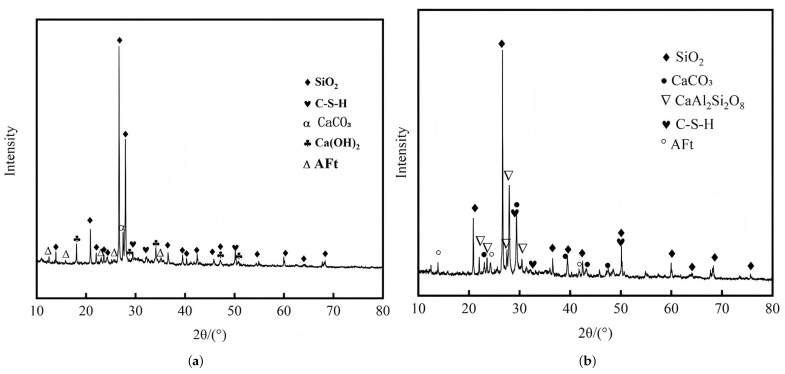
Comparison of XRD before and after dry–wet cycle of sulfate attack: (**a**) XRD analysis of 28-day concrete with natural desert sand. (**b**) XRD pattern of natural desert sand concrete with 90 cycles of wet–dry cycles of sulfate erosion [120].

**Figure 10 materials-18-02531-f010:**
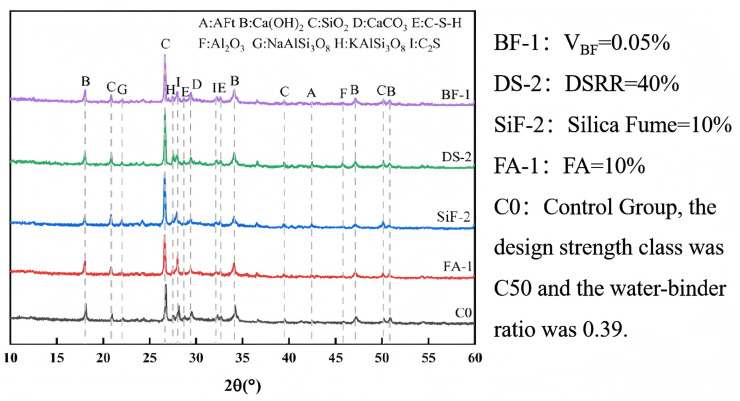
XRD patterns of several admixtures, strength class was C50, and the water-binder ratio was 0.39 [122].

**Figure 11 materials-18-02531-f011:**
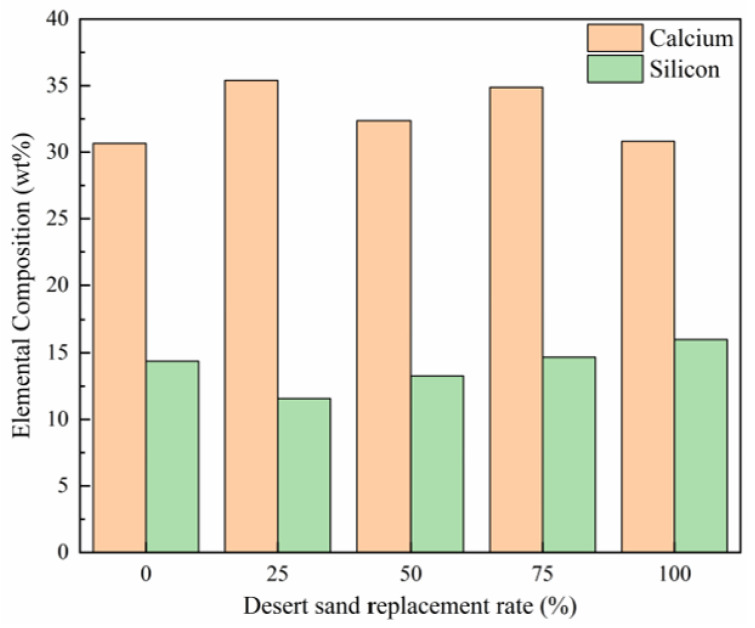
Variation in calcium and silicon composition with increasing CDS content [119].

**Figure 12 materials-18-02531-f012:**
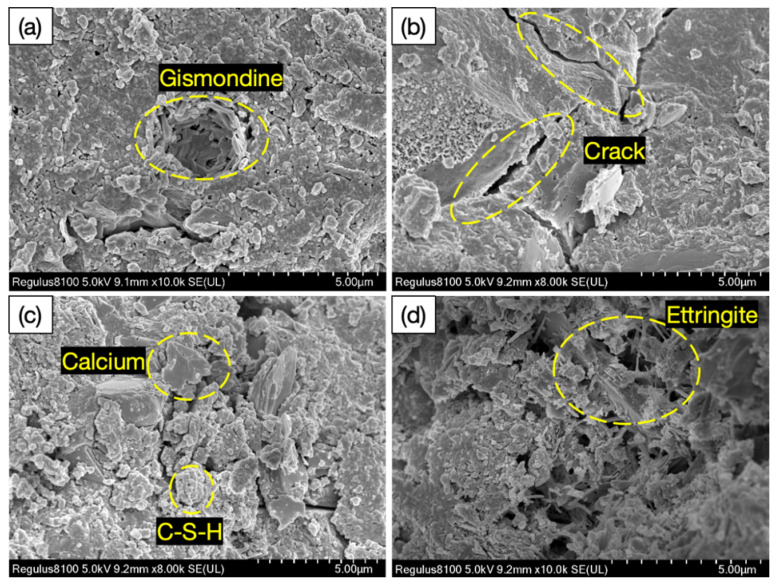
SEM images of glass fiber desert sand concrete hydration products: (**a**) Gismondine Crystal Aggregates. (**b**) Microcrack Formation in Matrix. (**c**) Calcium-Silicate-Hydrate (C-S-H) Network. (**d**) Ettringite Needle Clusters.

**Figure 13 materials-18-02531-f013:**
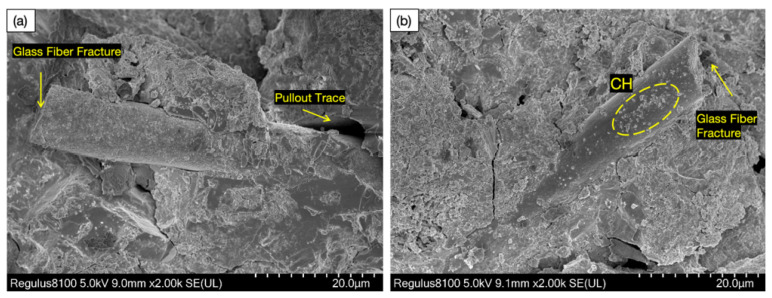
Fiber microstructure in glass fiber desert sand concrete: (**a**) Fiber Pull-out. (**b**) Fiber Fracture.

**Figure 14 materials-18-02531-f014:**
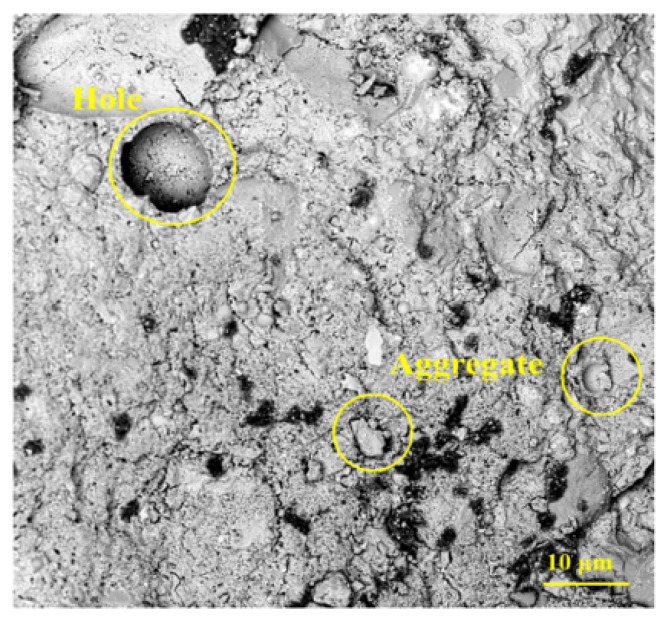
The SEM images of the ordinary DSC [84].

**Figure 15 materials-18-02531-f015:**
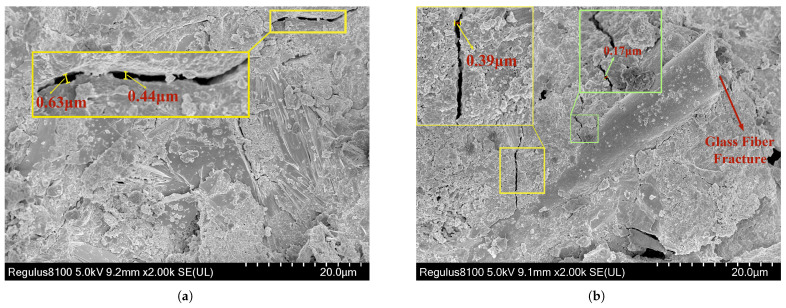
SEM comparison of cracks between pure matrix and fiber–matrix interface region: (**a**) SEM image of pure matrix interfacial crack; (**b**) SEM image of cracks at fiber–matrix interface.

**Table 1 materials-18-02531-t001:** Analysis of chemical composition of desert sand in different regions.

Source of Sand	Mass Fraction/%									Author
	**SiO_2_**	**Al_2_O_3_**	**CaO**	**Fe_2_O_3_**	**K_2_O**	**MgO**	**Na_2_O**	**TiO_2_**	**Other**	
River sand	97.53	2.84	0.00	0.19	0.76	0.00	0.00	—	—	S. Zhang [50]
Machine-made	73.59	7.59	3.07	4.83	1.33	1.02	1.75	—	6.82	S. Zhang [50]
Kubuqi	74.00	10.00	3.00	3.00	2.00	1.00	2.00	0.30	4.70	L. Wei [51]
Uulan Buh	69.00	10.00	5.00	3.00	3.00	2.00	3.00	0.20	4.80	R. Dong [52]
Tokxun	70.33	10.85	5.65	—	2.39	1.03	2.48	—	7.27	W. Huang [53]
Mu Us	82.66	8.72	2.00	—	0.12	1.51	0.07	—	4.92	H. Liu [54]
Tengri	82.92	8.02	0.22	—	0.06	1.39	0.03	—	7.36	A. Al-Harthy [35]
Gurbantunggut	63.62	9.63	8.19	2.15	2.08	2.14	2.83	0.22	9.00	G. Padmakumar [36]
Ausrtalia	94.8	2.00	0.23	0.66	0.34	0.11	0.06	—	1.8	F. Luo [39]
Takla Makan	55.61	9.56	14.38	2.44	2.32	2.54	2.08	0.36	10.41	W. Yan [37]

**Table 2 materials-18-02531-t002:** Physical and mechanical properties of different fibers.

Fiber Type	Diameter	Density	Elastic Modulus	Elongation at	Tensile Strength	Author
	(μm)	(g/${\mathrm{cm}}^{3}$)	(GPa)	Break (%)	(MPa)	
Steel Fiber (SF)	750	7.86	201	2.7	2850	H. Hamada [81]
Basalt Fiber (BF)	12	2.75	80–110	3.5	3000–4000	H. Hamada [81]
Polypropylene	31.86	0.91	>4.5	255±	567	Y. Tan [82]
Fiber (PPF)						
Polyvinyl Alcohol	15.3	1.3	40	7	1830	Z. Lina [83]
Fiber (PVAF)						
Glass Fiber (GF)	–	2.4	70	–	2500	L. Hou [84]
Carbon Fiber (CF)	7	–	240	1.5	4900	N. Feng [85]

**Table 3 materials-18-02531-t003:** Comparative analysis of the effects of different factors on mechanical properties.

Fiber Species and Volume Admixture	Desert Sand Substitution Rate (%)	Compressive Strength /MPa	Flexural Strength /MPa	Splitting Tensile Strength /MPa	Axial Compressive Strain /${\mathrm{10}}^{-3}$	Author
PPF 0.5%,	100	0.77	9.59	-	6.27	Y. Ma [114]
PVAF 1%						
PPF 0.1%,	30	45.7	4.10	4.38	-	S. Jian [112]
GF 0.05%						
PP 0.025%,	40	35.4	-	3.94	-	S. Sreebha [9]
BF 0.075%						
BF 0.1%	100	39.5	3.3	3.59	-	S. Lyu [117]
No added fiber	100	35.59	-	2.21	4.35	Z. Li [115]
No added fiber	60	33.5	-	1.57	2.24	Y. Ding [116]

**Table 4 materials-18-02531-t004:** Elemental compositions from EDS: data source [119].

Desert Sand Replacement Rate (%)	Ca (wt.%)	Si (wt.%)	Fe (wt.%)	Al (wt.%)	Mg (wt.%)	O (wt.%)
0	30.7	14.4	4.1	2.9	6.5	41.4
25	35.4	11.7	3.7	2.4	3.6	43.2
50	32.4	13.4	2.7	3.0	2.8	45.9
75	34.9	14.8	3.0	2.6	1.9	42.9
100	30.8	16.1	3.3	3.9	1.7	44.2

## Data Availability

The data presented in this study are available on request from the corresponding author due to privacy.

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
