# Peer review of "Research Status of Mechanical Properties and Microstructure of Fiber-Reinforced Desert Sand Concrete"

_materials, 2025, doi:10.3390/ma18112531_

Round 1

Reviewer 1 Report

Comments and Suggestions for Authors

The article briefly covers several topics related to desert sands, so the references used are not useful because the characterizing properties of the materials studied ("desert sands") are not specified in each study reported for each property explained in the article. Ideally, a review should elaborate on the explained properties, taking into account all the variables of the original material used ("desert sands"). As reported, these materials vary in their properties, so general behavior cannot be generalized for all sands. The reported properties are not related to the initial characteristics of the sands studied. There is no general discussion of the behavior of "desert sands" that relates all the reported properties.

Author Response

Thank you for offering us an opportunity to improve the quality of our submitted manuscript

materials-36150856. We appreciated very much the reviewers’ constructive and insightful comments. In this revision, we have thoroughly reviewed and ensured the relevance of all cited references to the manuscript, and have addressed all reviewer comments point by point. We sincerely hope that the revised manuscript now meets the publication standards of your esteemed journal.

We highlighted all the revisions in red color.

On the next pages, our point-to-point responses to the queries raised by the reviewers are listed.

We have carefully revised this manuscript according to your suggestions. And the modified part is marked in red.

Comment 1. The article briefly covers several topics related to desert sands, so the references used are not useful because the characterizing properties of the materials studied ("desert sands") are not specified in each study reported for each property explained in the article. Ideally, a review should elaborate on the explained properties, taking into account all the variables of the original material used ("desert sands"). As reported, these materials vary in their properties, so general behavior cannot be generalized for all sands. The reported properties are not related to the initial characteristics of the sands studied. There is no general discussion of the behavior of "desert sands" that relates all the reported properties.

Response: Thanks very much for the kind suggestions. We fully agree that the physical and chemical properties of desert sand vary depending on its origin, and these intrinsic characteristics have a critical impact on the performance of concrete. Therefore, it is essential to explicitly address this influence in the review. Our manuscript focuses on two central research themes: (1) the performance of desert sand concrete (DSC); and (2) the effects of various types of fibers on the mechanical properties of DSC. In response to your suggestions, we have revised and enriched the relevant content as follows:

  1. We conducted a more in-depth review of this topic and incorporated six additional references related to desert sand concrete [20][38][46][67][68][118] to enhance the comprehensiveness and currency of the literature review.
  2. In Section 2, “Basic Properties of Desert Sand,” we added more detailed descriptions of the physical and chemical characteristics associated with sand from different regions. In lines 105-115 of the revised manuscript, we included a new paragraph noting that desert sand from different regions exhibits certain similarities in chemical composition. Total dissolved solids (TDS) tests showed extremely low concentrations of sulfates and chlorides, suggesting negligible influence on the overall performance of the specimens. Furthermore, X-ray diffraction (XRD) analyses of desert sand from various regions confirmed mineral compositions similar to those of river sand. These findings have been incorporated and are supported by references [44][45].
  3. In Sections 3 and 4, which discuss mechanical properties and fiber reinforcement effects, we added specific descriptions of the desert sand sources in the cited studies. These additions, found in lines 152-153, 166-170, 193-194, and 364-372 of the revised manuscript, aim to improve the contextual relevance and rigor of our analysis.
  4. In the summary of Section 4.4 (lines 394-405), we introduced a novel perspective by analyzing the synergistic effect between the desert sand replacement rate and fiber content on enhancing the compressive strength of concrete specimens. This approach provides further insight into the combined action mechanisms of desert sand and fibers. Additionally, we suggest that future research should place greater emphasis on the standardization of desert sand characterization methods to ensure improved reproducibility and applicability in engineering practices.

Finally, we appreciate very much for your time in editing our manuscript and the referees for

their valuable suggestions and comments. We believe these revisions have enhanced the systematic structure and scientific depth of the manuscript, and we sincerely hope they meet with your approval.

Reviewer 2 Report

Comments and Suggestions for Authors

Due to the high rate of construction in desert regions, a significant amount of sand is required. In this regard, the topic under consideration is particularly relevant. The material is well structured, illustrations and tables are presented in high quality, and do not require additional processing.

The list of literary sources used is quite a wide selection, in the review I additionally recommended sources on the research topic for 2024-2025. Perhaps the authors will be able to add more references for this period. The novelty of the subject of the study lies in the use of fillers in the composition of concrete with desert sand, namely steel, polypropylene and basalt fibers. The novelty of the subject of the study is the use of fillers in the composition of concrete with desert sand, namely steel, polypropylene and basalt fibers. The complex use of these fibers increases the strength characteristics of concrete from desert sand, which makes it possible to widely use it in construction.

I believe that the article is written at a high level and requires minimal revision.

Dear authors, the material on the topic has been developed in sufficient detail using current literary sources. I suggest supplementing the article with the following links:

- DOI: 10.1016/j.rineng.2024.102478
- https://doi.org/10.1016/j.resconrec.2024.108002
- DOI: https://doi.org/10.32047/CWB.2024.29.3.2

Author Response

Dear Reviewer,

Thank you for your valuable and thoughtful feedback on our manuscript titled “Research status of Mechanical Properties and Microstructure of Fiber-Reinforced Desert Sand Concrete”. We greatly appreciate your thorough analysis, which has contributed significantly to improving the quality and clarity of our work.

We have made the following revisions to the manuscript:

  1. Thank you very much for taking the time to carefully review our manuscript and for recommending highly valuable references. We have incorporated the suggested references [20], [67], and [68] into the revised manuscript at lines 178–202 and 39.
  2. In response to comments from other reviewers, we have also expanded Section “3.2 Mechanical Properties of Desert Sand Concrete” to provide a more detailed description of the modeling process using ANSYS and ABAQUS software, including the specific failure criteria and boundary conditions employed in the simulations.
  3. Additionally, we carefully re-examined the manuscript and have provided supplementary explanations for figures and tables where the sources of data or information were previously unclear.

We sincerely thank you again for your thoughtful feedback. We believe these revisions have

improved the manuscript, and we hope that the revised version will meet your expectations.

Sincerely,

First author:

Dr. Bo Nan

Reviewer 3 Report

Comments and Suggestions for Authors

The reviewed article by Bo Nan , Jiantong Xin and Wei Yu entitled "Research status of Mechanical Properties and Microstructure of
Fiber-Reinforced Desert Sand Concrete" discussed the recent status of research on the mechanical properties and microstructure of
fiber-reinforced desert sand concrete. The manuscript consists of seven sections, where the properties of desert sand as well as different reinforcements ( steel fibers, polypropylene fibers, basalt fibers) are described.  The manuscript consists of 120 references.

The review is well prepared, however, several problems need to be corrected, namely:

1. Authors say that some mechanical properties are modeled using FEM (ANSYS, line 144, and Abaqus, line 203). However, there are no details of what kind of simulations are performed.

2. There is no information on how the DSC material is modeled and what kind of failure criteria are applied.

3. It is not clear if the following figures are made by the authors, namely: 7, 8, 12,13, 15, and Table 4.

4. There is no clear explanation of what "wear resistance of fly ash".

Author Response

Thank you for your approval. We have carefully revised this manuscript according to your suggestions. And the modified part is marked in red.

Comment 1. Authors say that some mechanical properties are modeled using FEM (ANSYS, line 144, and Abaqus, line 203). However, there are no details of what kind of simulations are performed.

Response: Thanks very much for the kind suggestions. We have carefully reviewed the manuscript and added more detailed information regarding the simulation process in lines 152-162 of the revised version. “H. Liu [62] used ANSYS software to establish a finite element model of Mu Us Desert sand concrete and performed a simulation of dynamic impact compression tests in the DSC”. The revised manuscript now includes a description of the specimen geometry, the use of the HJC constitutive model, and the meshing strategy.  Additionally, in lines 246-257 of the revised manuscript, we have expanded and clarified the simulation procedures conducted using ABAQUS. This includes detailed descriptions of the selected constitutive model, element types, boundary conditions, and the applied cyclic displacement loading protocol. These improvements aim to enhance the reproducibility of the modeling and the reliability of the simulation results.

Comment 2. There is no information on how the DSC material is modeled and what kind of failure criteria are applied.

Response: Thanks very much for the kind suggestions. We have thoroughly reviewed our manuscript and added detailed information in the revised version at lines 153-163 regarding the modeling process conducted using ANSYS software, including the specific failure criterion adopted. Additionally, the constitutive model equations employed in the Abaqus simulations have been included in the revised manuscript at lines 248-251 and lines 255-257.

Comment 3. It is not clear if the following figures are made by the authors, namely: 7, 8, 12,13, 15, and Table 4.

Response: Thanks very much for the kind suggestions. In the revised manuscript, Figures 7 and 8 have been renumbered as Figures 8 and 9, respectively. Figure 8 contains data sourced from references [84], [100], and [127]. We recompiled and appropriately modified the figure, and the respective references have been clearly indicated in the figure caption. Figure 9 is based on data from reference [113], which has also been appropriately edited and clearly cited in the caption. Figures 12, 13, and 15 have been renumbered as Figures 13, 14, and 16, respectively, in the revised manuscript. These figures were generated based on the authors original experimental images and results. To further clarify, we have added a note in lines 553-554 of the revised manuscript describing the materials used in the specimens shown in these figures. As for Table 4, it was created by the authors based on data from reference [116]. A note explaining the data source has now been added in the title row of the table in the revised manuscript for transparency.

Comment 4. There is no clear explanation of what "wear resistance of fly ash".

Response: Thanks very much for the kind suggestions. We have reevaluated the accuracy of the statement "wear resistance of fly ash" and made revisions in lines 501-513 of the revised manuscript. The updated statement is as follows "Experimental research by A. Cao [119] indicated that the abrasion resistance of concrete samples increased by 80.19% and 81.59% when incorporating 10% FA or 0.05% BF individually, respectively, while the addition of 10% silica fume (SiF) resulted in a 12.50% increase in compressive strength". Additionally, in lines 501-510, we explained the concepts of concrete wear resistance, fly ash and silica fume, and included a new references [118].